# Effects of Alternate Wet and Dry Conditions on the Mechanical and Physical Performance of Limestone Calcined Clay Cement Mortars Immersed in Sodium Sulfate Media

**DOI:** 10.3390/ma15248935

**Published:** 2022-12-14

**Authors:** Vincent Odhiambo Odhiambo, Lenka Scheinherrová, Silvester Ochieng Abuodha, John Nyiro Mwero, Joseph Mwiti Marangu

**Affiliations:** 1Department of Civil and Construction Engineering, University of Nairobi, Nairobi 00100, Kenya; 2Department of Materials Engineering and Chemistry, Faculty of Civil Engineering, Czech Technical University in Prague, Thákurova 7, 166 29 Prague, Czech Republic; 3Department of Physical Sciences, Meru University of Science and Technology, Meru 60200, Kenya

**Keywords:** compressive strength, ordinary Portland cement, durability, sorptivity, supplementary cementitious materials, calcined clay

## Abstract

Sulfate attack in concrete structures significantly reduces their durability. This article reports the experimental findings on the effects of sodium sulfate on limestone calcined clay cement (LC^3^) in an alternate wet and dry media. The samples underwent wet–dry conditions of 28 cycles. Two types of LC^3^ were studied, one made from clay (LC^3^-CL) and the other made from fired rejected clay bricks (LC^3^-FR). The composition of each LC^3^ blend by weight was 50% clinker, 30% calcined clay, 15% limestone, and 5% gypsum. The reference compressive strength was evaluated at 2, 7, and 28 days of age. Then, ordinary Portland cement (OPC) and LC^3^-CL blends were subjected to alternate wet–dry cycle tests, immersion in a 5% sodium sulfate solution, or in water. For all exposed samples, sorptivity tests and compressive strength were done. The results showed that LC^3^ blends met the requirements for KS-EAS 18-1:2017 standard, which specifies the composition and conformity criteria for common cements in Kenya. The LC^3^ blend also had a lower rate of initial absorption compared to OPC. Additionally, LC^3^ blend also showed good resistance to sodium sulfate when exposed to alternating wetting and drying environment. OPC showed higher compressive strength than LC^3^ blends for testing ages of 2, 7, and 28 days. However, the LC^3^ samples utilized in the sodium sulfate attack experiment, which were later tested after 84 days, exhibited higher compressive strengths than OPC tested after the same period.

## 1. Introduction

Concrete is the most widely used construction material worldwide [1]. It is used in the construction of buildings, bridges, road pavements, dams, and railway lines, among others. Cement is a major component of concrete production. There are two main types of cement used in the manufacture of concrete, namely ordinary Portland cement (OPC) and Portland pozzolana cement (PPC). OPC is made up of clinker and gypsum, while in the PPC, the clinker is partially replaced by a supplementary cementitious material (SCM) [2]. OPC is an unaffordable binder in developing countries, mainly due to the high energy requirements in the production of clinker. Additionally, the production of OPC accounts for about 5–8% emissions of carbon dioxide (CO_2_) worldwide [2]. CO_2_ emissions are mainly responsible for global warming and consequently lead to climate change [3].

Partial replacement of clinker with SCMs is an innovative strategy to lower the cost of cement and reduce the CO_2_ emission [4]. Most recently, calcined clay and limestone have been combined as SCMs in cement production. LC^3^, as this solution is called, has an advantage in that both limestone and clay are readily available in large quantities. The calcination processes of clay also require less energy than those of clinker, leading to reduced costs of resources such as those of electricity. As opposed to clinker, the calcination process does not result in the production of carbon dioxide from the breakdown of the raw clays; hence, it helps in the reduction of CO_2_ emission [5]. Recently, Li et al. [6] reported that the incorporation of 15–30% calcined cutter soil mixing residue helped to reduce both the embodied energy and CO_2_ emissions of cement mortars by around 13–25%.

Cement structures are susceptible to degradation when exposed to sulfate media, which is usually encountered in construction environments [7,8,9]. Sources of sulfate ions that degrade concrete structures can be found in soils, seawater, sewerage, and effluence from factories, among other sources [10,11]. Sulfate attack typically leads to cement mortar degradation. Degradation of concrete structures as a result of sulfate attack can manifest in various forms. Some of these forms include expansion, cracking, softening, spalling, as well as strength loss [12]. Sulfate attacks can either be physical or chemical in nature. The physical attack mainly occurs in alternating wet and dry conditions. This occurs because of the precipitation of sulfate salts in the pores of the cement specimen structure. This type of attack is common with structures in tidal areas, such as piers supporting sea-crossing bridges and port docks that are within splash zones, sewerage ducts, irrigation channels, etc. Half-buried or submerged structures can also experience this cyclic wetting and drying and are thus prone to sulfate attack [13].

Portland Cement Association, recognized as PCA, has carried out research that showed that a physical sulfate attack is more severe than a chemical attack [14]. A chemical sulfate attack involves a reaction between hydration products and sulfate ions to form ettringite and gypsum. The sulfate ions require aluminum and calcium to form ettringite. Ettringite is an expansive crystalline product, and it occupies a larger volume than the products that form it. The expansive nature of ettringite may result in cracks if the tensile strength of cement is exceeded as a result of outward pressure exerted on the pore walls [15]. It is possible for both chemical and physical attacks to coincide in a single specimen.

Various factors contribute to the severity of sulfate attack. El-Hachem et al. [16] noted that the beginning of specimen expansion depends on the size of the specimen and that this could be delayed by an increase in the size of the specimen. The condition of curing also affects the degree of sulfate damage. Whittaker and Black [17] noted that samples cured in the air have better sulfate resistance due to hindered sulfate ion transportation due to the formed carbonation layer compared to those cured in water. They also noted that the concentration of sulfate solution does not significantly affect the depth of propagation in mortar specimens. Permeability can enhance sulfate ion transportation and thus enhance sulfate attack. Highly permeable cement mortars are, therefore, highly likely to be more vulnerable to sulfate attack compared to those that are less permeable. High concentrations of sulfate ions can speed up the deterioration and lead to the production of gypsum.

The type of sulfate source also plays a role in the severity of the attack. Bescher et al. [12] indicated that magnesium sulfate had the most severe effect, followed by sodium sulfate and, lastly, calcium sulfate. The mechanism for sodium sulfate attack may involve single or multiple reactions to form ettringite. In the single reaction, tricalcium aluminate (C_3_A) reacts with sodium sulfate to form ettringite. In the multiple reactions, sodium sulfate and portlandite formed from cement hydration interact together. The reaction results in the formation of gypsum, which in turn reacts with sodium sulfate to form ettringite.

One of the methods that have been proposed to reduce sulfate attack in cement mixtures is the prevention of the expansive sulfate attack products. This can be controlled by reducing the amount of C_3_A and portlandite, which are the primary sources for the formation of ettringite. Whittaker and Black [17] in their review reported improved sulfate resistance with lower content of tricalcium aluminate. Partial replacement of cement clinker with SCMs having high silica content may also help improve sulfate resistance. SCMs reduce the formation of ettringite and gypsum since they react with (i.e., consume) portlandite formed during the hydration of cement. This decreases the amount of portlandite available for ettringite and gypsum formation [10]. The addition of SCMs also results in a dense structure with a lower volume of pores compared to OPC, further improving resistance to salts such as sodium sulfate. Calcined clays used in the design of LC^3^ usually have a high silica content, which also helps in the prevention of the development of expansive sulfate attack products.

This study evaluated the effects of alternate wet and dry conditions on the mechanical performance of LC^3^ mortars immersed in sodium sulfate media. This was done by comparing the compressive strength test and sorptivity test results of samples subjected to alternating wet–dry conditions, immersion in sodium sulfate, or in water. Additionally, this study evaluated the mechanical performance of LC^3^ made from clay soil and those made from fired rejected clay bricks (FRBCs) to determine if there was any significant difference in their compressive strength performance. The compressive strength results at 2, 7, and 28 days of curing in water were compared with the specifications in KS EAS 148-1:2017 cement standard [18].

## 2. Materials and Methods

### 2.1. Raw Materials

Clay was sampled from Mwea in Kirinyaga County, while FRCBs were obtained from a building demolition site in Nairobi, more specifically from calcined clay roofing tiles. The LC^3^ samples were labeled as follows: LC^3^-CL (Mwea clay) and LC^3^-FR (FRCBs sample). Commercially obtained PPC conforming to 32.5 MPa and OPC conforming to 42.5 MPa were used as reference cements and compared with the newly designed LC^3^. The properties of raw materials are shown in Table 1. For the durability test, 5% Na_2_SO_4_ was used to evaluate the performance of LC^3^ under sulfate attack.

### 2.2. Sample Preparation

The used clay soil was calcined in a furnace (Model 10-D1418/A) at the temperature of 800 °C for 1 h. The calcined clays were then left to naturally cool down to room temperature and then pulverized with a target residue of 30% when passed through a 45-micron sieve. The FRCBs sample was also subjected to the same heat treatment as the clay sample. The drying of the sample was needed to remove any moisture that might have accumulated over time. The LC^3^ samples were then prepared by mechanical mixing of 50 wt.% of clinker, 30 wt.% of calcined clay, 15 wt.% of limestone, and 5 wt.% of gypsum. Mortar prisms were prepared and cured in accordance with KS EAS 148-1:2017 standard [18], specifying the composition and conformity criteria for common cements in Kenya. This involved mixing of 450 g of the cementitious part (OPC; LC^3^-CL, or LC^3^-FR), 1350 g of standard sand, and 225 mL of water. At first, the dry mixture was added to the desired amount of water and mixed in an automatic mixer for ten minutes. The fresh mixture was then placed in a mold with 40 × 40 × 160 mm dimensions and then vibrated using a vibration compaction machine at a speed of 30 blows per minute. The fresh samples were kept in a temperature-controlled room with a humidity level above 90% and a temperature of 21 °C. Mortars were de-molded after 24 h and were placed in a curing water tank for 2, 7, and 28 days.

### 2.3. Methods

#### 2.3.1. Compressive Strength

For the first part of the experiment, which included determining the compressive strengths of studied samples for 2, 7, and 28 days of water curing, a total of 9 prisms were prepared. At each testing age, 3 mortar prisms from each sample set were taken out of the curing tank, wiped, and given 10 min to drain. They were then placed in a compressive strength test machine model YAW-300 in order to determine their compressive strength. The compressive strength results were recorded in MPa.

For the second part of the experiment, which involved exposing the cement samples to sodium sulfate (it is described in detail in Section 2.3.2), a total of 9 prisms were cast for each sample set. The compressive strength after the exposure experiments (at 84 days of age) was tested.

#### 2.3.2. Sulfate Attack

The OPC and LC^3^-CL samples were subjected to a sulfate attack test. Firstly, the prisms were cast and then cured for 28 days in water in accordance with KS EAS 148-1:2017 [18]. After 28 days of curing, the cement prisms were kept in the furnace for 5 days (until a constant weight was achieved) at a temperature of 50 °C to stop hydration and to keep the same moisture level for all analyzed samples. The samples were then stored at room temperature for 10 days before being exposed to a 5% sodium sulfate solution. For each mixture, 3 sets, each consisting of 3 prisms, were prepared. One set underwent a wet–dry cycle test for 28 days (6 h of wetting and 18 h of drying in a furnace at 60 °C per cycle = 28 cycles in total). These samples were labeled as OPC-WD and LC^3^-CLWD. Another set was exposed to a water environment again for another 28 days. These prisms were named OPC-RW and LC^3^-CLRW. Finally, the last one was immersed in a 5% sodium sulfate solution for 28 days, and these samples were labeled as OPC-RS and LC^3^-CLRS. Table 2 summarizes the labeling of the samples.

After completion of the sulfate attack experiment, these samples underwent a sorptivity test for another 12 days after which they were subjected to a compressive strength test. A total of 84 days passed from the day these samples were cured to the day they were subjected to compressive strength testing.

#### 2.3.3. Sorptivity Test

ASTM C 1585-13 method [19] was used to conduct the sorptivity test with slight modifications: the cement prisms, which were previously subjected to the wet–dry cycle test, and those recurred in water and in 5% sodium sulfate solution, were subjected to this test to evaluate their water absorption characteristics. The samples were dried in a furnace for 3 days at 50 °C until a constant weight was achieved. They were then placed in a sealable container for 10 days to allow for equilibration of the moisture distribution within the test specimens. Each sample was then submerged in water at a level of 3 mm from the bottom in accordance with the setup in ASTM C 1585-13 [19]. After each exposure interval, the weights of the samples were recorded in grams after the submerged surface was dried with a clean, damp towel. The exposure intervals were 1, 4, 9, 16, 25, 36, 49, 64, 81, 100, 121, 169, 225, 289, and 361 s. Absorption was calculated using Equation (1) [19].
*I* = Δm_t_/(a × d).(1)

The absorption, *I* (in mm), is the change in weight divided by the product of the cross-sectional area of the test specimen and the density of water. More specifically, Δm_t_ is the change in weight of the sample in grams, a is the exposed area of the specimen (mm^2^), and d is the density of the water in g/mm^3^ (taken as 0.001 g/mm^3^). The initial rate of water absorption (mm/√s) is the slope of the line that is the best fit to I plotted against the square root of time. This slope is obtained by using least-squares, linear regression analysis. For the regression analysis, all the points from 1 min to 6 h are used, excluding those points for times after the plot shows an apparent change of slope.

## 3. Results and Discussion

### 3.1. Compressive Strength of Water-Cured Samples

The results of the compressive strength for the LC^3^, PPC, and OPC samples after 2, 7, and 28 days of curing in water are shown in Figure 1.

As expected, the compressive strength increased with the curing period for all cement categories. This is attributed to the process of cement hydration of the main cement phases, such as C_3_S (Ca_3_SiO_5_), C_2_S (Ca_2_SiO_4_), C_3_A (Ca_3_Al_2_O_6_), and C_4_AF (Ca_4_Al_n_Fe_2-n_O_7_), which leads to progressive strength gain with curing time [19]. Hydration of C_3_S and C_2_S are the main phases responsible for strength development. The reaction of these main phases with water (H_2_O) can be represented by Equations (2) and (3) [20,21].
2Ca_3_SiO_5_ + 6H_2_O → Ca_3_Si_2_O_7_·3H_2_O + 3Ca(OH)_2_,(2)
2Ca_2_SiO_4_ + 4H_2_O → Ca_3_Si_2_O_7_·3H_2_O + Ca(OH)_2_.(3)

As CH (portlandite) and mainly amorphous C-S-H phase are formed, the cement begins to harden. The C-S-H phase is a cementitious material responsible for the binding of cement to other concrete constituents. The process of hydration continuously consumes cement and water. This is why water curing is essential; because, without water, the process will not continue. The reaction products, C-S-H and portlandite, occupy almost equal volume as the reactants, and hence volume is almost conserved, and shrinkage remains within a manageable range [21].

In blended cements, portlandite released during hydration reacts with silica (SiO_2_) and alumina (Al_2_O_3_) from calcined clays to form an additional C-S-H phase and also the C-A-H phase. These reactions can be summarized as shown in Equations (4) and (5) [20].
3Ca(OH)_2_ + 2SiO_2_ + 3H_2_O → 3CaO·2SiO_2_·6H_2_O,(4)
Al_2_O_3_ + H_2_O + Ca(OH)_2_ → CaO·Al_2_O_3_·2H_2_O.(5)

The LC^3^ also benefits from the synergetic effect of limestone reaction with additional alumina provided by calcined clay. This makes it possible to maintain good mechanical performance and durability at higher levels of substitution [22,23]. Nied et al. [24] also observed that the use of calcined clay (metakaolin) and limestone together in cement positively affects both compressive strength and workability.

In OPC, the hydration of C_3_S is responsible for most of the early strength in cement mortars, while later strength gains are mainly attributed to the hydration of the C_2_S [21]. Since the OPC samples contained a higher proportion of C_3_S and C_2_S, they exhibited higher compressive strength than all blended cement samples within the first 28 days of curing [20]. When compared numerically, the results obtained in this study were similar to those reported by Marangu [25]. Lavanya and Rao [26] also tested concrete samples using OPC and LC^3^ and obtained similar results. Both LC^3^ samples made from clay soil and the ones made from FRCBs met the compressive strength requirements of the KS-EAS 18-1:2017 [18] standard. For 2 days, the standard specifies a compressive strength of ≥10 MPa for PPC and ≥20 MPa for OPC. The LC^3^-CL and LC^3^-FR had a 2-day compressive strength of 14.7 MPa and 14.1 MPa, respectively. For 7 days, the standard specifies a compressive strength of ≥16 MPa for PPC but does not provide any value for OPC. The LC^3^-CL and LC^3^-FR had a 7-day compressive strength of 32.6 MPa and 31.5 MPa, respectively. Finally, for 28 days, the standard defines a compressive strength of ≥32.5 MPa for PPC and ≥42.5 MPa for OPC. The studied LC^3^-CL and LC^3^-FR exhibited a 28-day compressive strength of 44.0 MPa and 45.4 MPa, respectively. This indicates that Kenya has the potential to produce LC^3^ using both clays and FRCBs, as they both meet the standard’s specification for compressive strength.

### 3.2. Compressive Strength after Sulfate Attack

Figure 2 shows the compressive strength results for LC^3^ and OPC, which were subjected to wet–dry cycle testing and recurred in water and 5% sodium sulfate. From Figure 1, it is visible that after 28 days of curing in water, the OPC samples exhibited higher compressive strength than the LC^3^ samples. This, however, is not the case in Figure 2, which shows that the LC^3^ samples after 84 days had significantly higher compressive strength than the OPC samples. This phenomenon is a result of the secondary reactions that happen when blended cements, such as LC^3^, are used. During this reaction, portlandite which was formed during the hydration of calcium silicates transforms into the C-S-H phase and C-A-H phase through pozzolanic reactions with the SCMs in the presence of water [27,28].

Both the OPC and LC^3^ samples that were subjected to wet–dry cycles exhibited the lowest strength in all three sets. The percentage loss of strength is shown in Table 3. Sulfate salts usually degrade porous structures by either physical or chemical processes. The loss in strength in both OPC and LC^3^ samples can be attributed to both physical and chemical sulfate attacks.

The physical damage takes the form of precipitation of sulfate salts in the pores of the structure. Alternating wet and dry conditions can favor such damages. In Na_2_SO_4_-H_2_O, there exist two stable phases thenardite (Na_2_SO_4_), which is the anhydrous and mirabilite (Na_2_SO_4_.10H_2_O) phase [29]. Sodium sulfate undergoes a significant volume change when it converts from anhydrous (thenardite) to a hydrous (mirabilite) phase. This can happen at temperatures below 32 °C if a sample that contains thenardite is exposed to increasing humidity [30]. Whittaker and Black [17] in their review indicated that at humidities above 75% and temperatures below 35 °C, a solution of thenardite is supersaturated with respect to mirabilite. This results in the precipitation of mirabilite. At temperatures above 32.4 °C, thenardite is reported to precipitate directly from the solution [29].

Damage (which might result in strength loss) occurs when stresses caused by crystallization exceed the tensile strength of the porous material. Studies have reported that the crystallization stresses resulting from the crystallization of sodium sulfate are higher than the tensile stresses of most stones [30,31]. SCMs such as calcined clay, when used in cement, may help in the prevention or delay of sulfate attack [32]. This, however, is not the case with physical sulfate attack. In fact, the use of SCMs may aggravate the situation more as their use leads to an increase in pores of smaller diameter. Nehdi et al. [33] investigated sulfate attack in concrete and reported that concrete containing pozzolans experiences more significant damage from physical sulfate attack than OPC. They attributed this to increased pores of smaller diameter, which in turn increased capillary suction and the surface area for drying. This explains the loss in strength in the LC^3^ sample that was subjected to wet–dry cycles.

The LC^3^ samples immersed in a 5% sodium sulfate solution also had a lower compressive strength than the reference LC^3^ samples that recurred in water. These samples are expected to undergo mostly chemical sulfate attack since they were continuously submerged in the solution. A chemical sulfate attack results from the contact between sulfate ions and the cement paste [11]. This requires the transfer of sulfate ions from the surface into the concrete structure for major deterioration to occur. This is normally facilitated by a concentration gradient and can be inhibited by the permeability of the structure.

Chemical attack can result from either the precipitation of gypsum and ettringite or the leaching of calcium hydroxide out of cement paste. Ettringite precipitation may lead to expansion and cracking. Its formation involves the reaction of sulfate ions with calcium and aluminum. Ettringite formation is associated with an overall loss of volume and, therefore, is not expansive itself. However, this conversion increases by more than double the total volume of solids formed, which, as a result, may cause expansion and cracking. This is also the case with gypsum formation from portlandite. Eglinton [34] reported an increase of 33.2 mL/mol to 74.1 mL/mol in the case of gypsum, while in [17], a rise of 312.7 mL/mol to 714.9 mL/mol was reported in the case of ettringite.

Baghabra Al-Amoudi [35] suggested that blended cements were more resistant to sulfate attack because they consume portlandite during pozzolanic reactions, which, as a consequence, leads to the denser microstructure of the hardened cement paste matrix. The dense structure inhibits sulfate ion transfer, while the consumption of portlandite reduces the amount of calcium, therefore leading to reduced ettringite formation. The use of blended cement also reduces tricalcium aluminate and tricalcium silicate in cements. This results in less portlandite and aluminate hydrates available to react with sulfate ions. This also contributes to an improved resistance of blended cements to chemical sulfate attack.

Even though the LC^3^ samples that were immersed in a 5% sodium sulfate solution showed a lower strength than the ones that recurred in water, the reduction in strength was negligible (3.6%). This is an indication that the LC^3^ exhibited good sulfate resistance. The slight fall in strength could also be caused by the fact that these samples did not purely undergo a chemical sulfate attack. During the sorptivity test, part of the samples was submerged in water while the other portions were exposed to air. This created a situation similar to that of a wet–dry cycle, which might have facilitated physical sulfate attack. Parallels can be drawn between this work and studies carried out by Scherer [31], who partially submerged porous stones in a sodium sulfate solution in his experiments. The sodium sulfate solution penetrated the stone sample by capillary action, which later caused efflorescence on the sides of the sample. The LC^3^ samples immersed in a 5% sodium sulfate solution also experienced slight efflorescence during the sorptivity test, but this phenomenon was more pronounced in samples that had undergone wet–dry cycles.

The OPC samples immersed in a 5% Na_2_SO_4_ solution had higher strength than the OPC samples recurred in water. These results are similar to those of Tian and Han [36]. They attributed the gain in strength to hydration-expansive products such as ettringite, gypsum, and sulfate crystals. These products fill the pores and, as a result, improve the specimen’s density and strength. However, they observed that prolonged exposure of the specimens to sulfate solution led to a reduction in strength. This is because the expansion force of gypsum and ettringite, as well as pressure as a result of sulfate crystallization, at some point, exceed the tensile strength of concrete. This causes internal microcracks, which in turn lowers the strength. Lv et al. [37], while using cement prisms, reported similar observations. They observed an initial gain in strength when the cement prisms were exposed to a 5% Na_2_SO_4_ solution for the first 3 months. The strength later decreased as it approached an exposure period of 12 months. They also attributed the initial increase in strength before deterioration to the formation of expansive hydration products.

### 3.3. Sorptivity Test

#### 3.3.1. Water-Cured Samples

Figure 3 shows the sorptivity test results for the samples that were recurred in water. From the results, it can be seen that LC^3^ had an initial absorption rate of 1.46 × 10^−2^ mm/√s, while the OPC samples exhibited a significantly higher initial absorption rate of 12.5 × 10^−2^ mm/√s. Blended cements such as LC^3^ and PPC typically have improved permeability resistance due to secondary (pozzolanic) hydration reactions [32].

This reduction in permeability typically increases the corrosion resistance in such cements [10]. Saraswathy et al. [38] found similar results when they tested OPC, PPC and Portland slag cement (PSC) samples. They observed as well that the blended cements (PPC and PSC) had a lower coefficient of water absorption than OPC. Dhandapani et al. [39] also found similar results.

#### 3.3.2. Samples Subjected to Wet–Dry Cycles

Figure 4 shows the sorptivity test results for samples subjected to wet–dry testing. Both LC^3^ and OPC samples subjected to wet–dry test exhibited higher initial rates of absorption compared to their counterparts that were cured in water. LC^3^ had an initial absorption rate of 5.85 × 10^−2^ mm/√s, while OPC had 15.19 × 10^−2^ mm/√s. This could be a result of internal cracks developing inside the samples subjected to wetting and drying conditions. The cracks can be a result of sulfate crystallization or varying hygrometric profile within the samples, causing shrinkage or both [13,36,40].

When compared to the LC^3^ and OPC samples subjected to the wet–dry test, the LC^3^ samples still showed a relatively lower initial rate of absorption than the OPC samples. This is an indication that LC^3^ could provide better resistance to ingress of salts such as sodium sulfate, hence better performance in such corrosive environments [10].

#### 3.3.3. Samples Immersed in a 5% Sodium Sulfate Solution

Figure 5 shows sorptivity test results for samples immersed in a 5% Na_2_SO_4_ solution. OPC immersed in a sodium sulfate solution had a lower initial rate of absorption (10.1 × 10^−2^ mm/√s) compared to similar OPC samples recurred in water (12.5 × 10^−2^ mm/√s). This can be attributed to the densification of OPC structure on early exposure to sodium sulfate. This densification is due to the formation of expansive hydration products such as ettringite, gypsum, and sulfate crystals [36]. This improvement can also be seen in the enhanced compressive strength values of the OPC sample immersed in sodium sulfate relative to samples that were cured in water as shown in Figure 2.

LC^3^ immersed in sodium sulfate exhibited a higher initial rate of absorption (9.7 × 10^−2^ mm/√s) compared to LC^3^ cured in water (1.46 × 10^−2^ mm/√s). This increase in initial absorption rate could be due to the increased number of pores in the structure of the LC^3^ samples. The pores might have occurred as a result of a chemical sulfate attack as well as a physical attack. The increased pores might also be responsible for the compressive strength reduction observed in the LC^3^ cured in sodium sulfate relative to LC^3^ samples cured in water, as seen in Figure 2. The LC^3^ recurred in sodium sulfate still recorded a lower initial rate of absorption compared to OPC samples subjected to the same conditions. This shows that LC^3^ is denser in its structure and will hinder penetration of ions, thereby improving its resistance to salts such as sodium sulfate [39].

## 4. Conclusions

In this study, the experimental findings on the effects of sodium sulfate on limestone calcined clay cement (LC^3^) in an alternate wet and dry media are presented. Based on the obtained results, the following conclusions can be made:From the results of compressive strength test carried out on samples exposed to sulfate attack and the reference samples, we can conclude that LC^3^ has good resistance to sodium sulfate attack. All the LC^3^ samples immersed in a 5% sodium sulfate solution and those that underwent wet–dry cycles attained a compressive strength of 61.3 ± 0.36 MPa and 57.7 ± 2.96 MPa, respectively. These strengths were not far off from the compressive strength of 63.6 ± 0.90 MPa, attained by the reference LC^3^ samples that were submerged in water.From the sorptivity test results, we can conclude that the use of calcined clay and limestone as SCMs in cement resulted in a denser structure. This was evidenced by the low initial rate of absorption exhibited by LC^3^ samples (1.46 × 10^−2^ mm/√s) compared to OPC samples (12.5 × 10^−2^ mm/√s) for samples cured in water.The designed LC^3^ showed good resistance to varying hygrometric conditions. When subjected to alternate wet and dry cycles, the LC^3^-CL samples attained a compressive strength of 57.7 ± 2.96 MPa upon testing. This strength is close to the reference LC^3^-CL sample which remained fully immersed in water that attained a compressive strength of 63.1 ± 0.90 MPa.There is potential for LC^3^ production in Kenya using clay soil and FRCBs. Chemical analysis showed that the composition of the clay soil and FRCB sampled satisfy requirements for a good pozzolan as prescribed in KS-EAS 18-1:2017 [18] standard. The standard specifies that silicon dioxide (SiO_2_) content shall not be less than 25% by mass of the material used. From Table 1, it is evident that both clay soil and FRCB samples meet this criterion. The clay soil sample had 39.02 wt.% of SiO_2_, while FRCB had 43.22 wt.%. The 2, 7, and 28-day compressive strengths also met the standard’s requirements of ≥10 MPa for PPC and ≥20 MPa for OPC for 2 days, ≥16 MPa for PPC for 7 days and ≥32.5 MPa for PPC and ≥42.5 MPa for OPC for 28 days.

## Figures and Tables

**Figure 1 materials-15-08935-f001:**
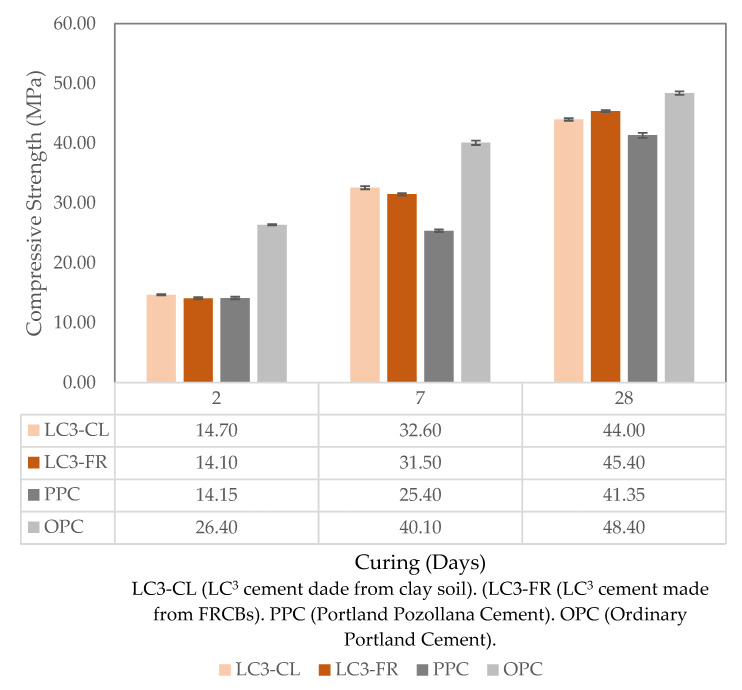
Compressive strength results for LC^3^ blends, PPC, and OPC.

**Figure 2 materials-15-08935-f002:**
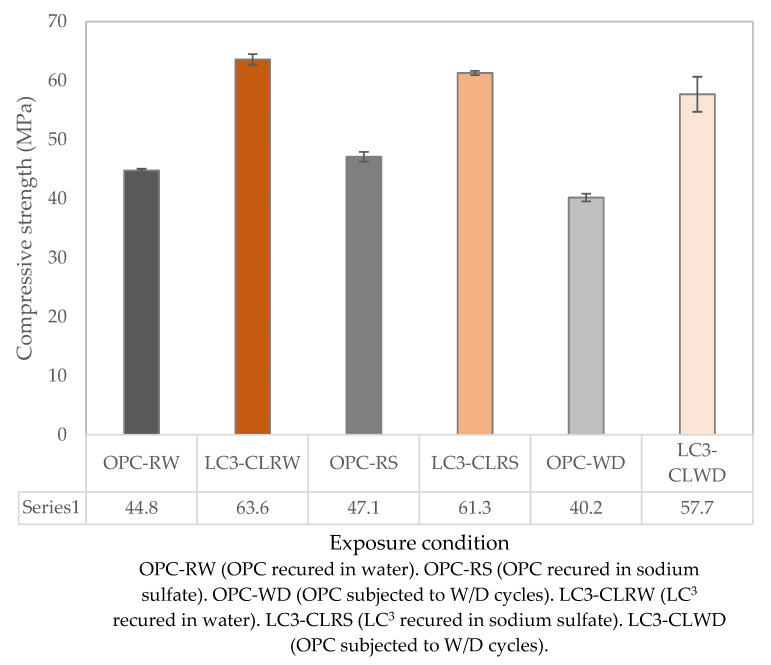
Compressive strength results for OPC and LC^3^-CL subjected to sulfate attack.

**Figure 3 materials-15-08935-f003:**
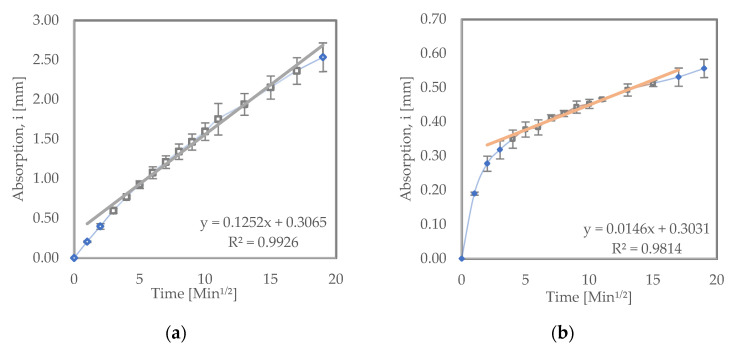
Sorptivity results for (**a**) OPC and (**b**) LC^3^ samples recurred in water.

**Figure 4 materials-15-08935-f004:**
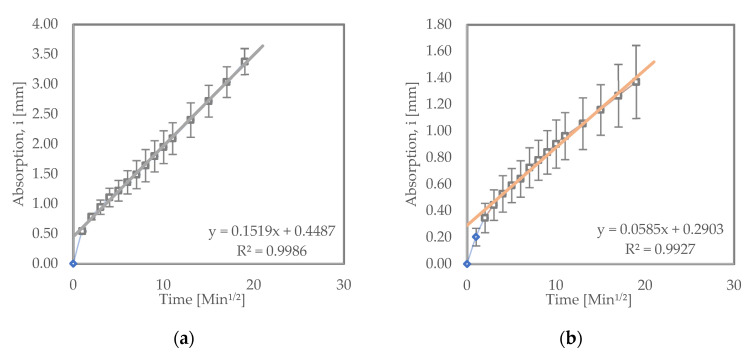
Sorptivity results for (**a**) OPC and (**b**) LC^3^ samples subjected to wet–dry cycles.

**Figure 5 materials-15-08935-f005:**
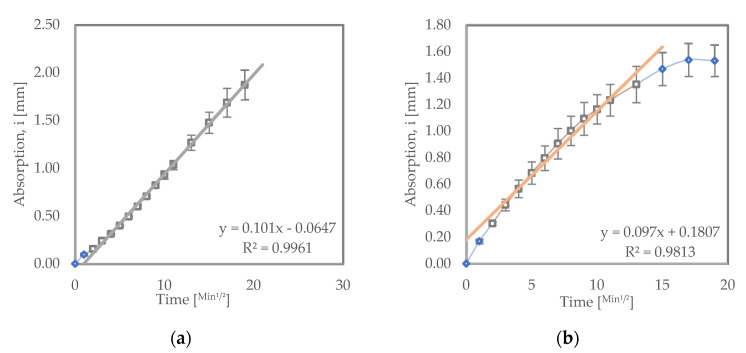
Sorptivity results for (**a**) OPC and (**b**) LC^3^ samples immersed in a 5% sodium sulfate solution.

**Table 1 materials-15-08935-t001:** Chemical composition (in wt.%) and selected physical properties of raw powders.

Composition	LC^3^-CL	LC^3^-FR	OPC	PPC	C *	G *	L *	CS *	FRCBs
SiO_2_	35.72	39.47	28.59	37.39	27.44	12.56	3.76	39.02	43.22
CaO	34.15	33.68	60.10	41.00	62.87	-	-	1.32	1.47
Al_2_O_3_	13.42	16.02	4.32	8.52	4.29	1.17	0.49	20.82	20.59
SO_3_	3.03	2.93	1.77	2.66	0.34	-	0.33	-	-
Fe_2_O_3_	10.64	6.37	4.51	7.13	4.43	1.64	0.52	15.18	7.96
TiO_2_	2.01	0.77	0.36	0.58	0.32	0.07	0.03	3.32	0.99
K_2_O	0.44	0.67	0.23	2.45	0.11	0.15	0.05	0.73	1.06
MnO	0.20	0.10	0.10	0.22	0.10	0.07	0.02	0.14	0.05
P_2_O_5_	0.39	-	-	-	0.10	-	-	0.45	-
LOI (wt.%)	7.84	5.73	3.43	3.60		-	-	1.67	1.28
Fineness (Residue in 45-micron sieve) (wt.%)	20.40	21.70	20.17	22.40	-	-	-	8.50	9.00
Insoluble Residue (wt.%)	12.30	29.34	1.05	24.00	-	-	-	50.21	67.06
Calculated surface (m^2^/cm^3^)	0.86	0.77	0.68	0.70	-	-	-	-	-

* C—Clinker, G—Gypsum (CaSO₄·1/2H₂O), L—Limestone, CS—Clay soil.

**Table 2 materials-15-08935-t002:** Labeling of the samples subjected to sulfate attack.

	Wet–Dry Cycle Test	Water Curing	5% Sodium Sulfate Solution
OPC	OPC-WD	OPC-RW	OPC-RS
LC^3^-CL	LC^3^-CLWD	LC^3^-CLRW	LC^3^-CLRS

**Table 3 materials-15-08935-t003:** Strength loss after wet–dry cycle test.

Sample	Reference Sample Compressive Strength	Compressive Strength After Wet–Dry Cycle Test	Strength Loss
LC^3^-CL	63.60 MPa	57.7 MPa	9%
OPC	44.80 MPa	40.20 MPa	10%

## Data Availability

Not applicable.

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
