# Peer review of "Effects of Alternate Wet and Dry Conditions on the Mechanical and Physical Performance of Limestone Calcined Clay Cement Mortars Immersed in Sodium Sulfate Media"

_materials, 2022, doi:10.3390/ma15248935_

Round 1

Reviewer 1 Report

1-The cement mix (in kg/m³ or % of the constituent materials-cement, clay, and ..) was not provided, which implies the impossibility of other authors replicating the study; therefore, I consider it important that these data are reported in the article.

2-  Carefully review the language, especially the punctuation, for example, sentence beginnings without capital letters.

3- Was any specific parameter in cement mortar configuration held constant for comparative purposes?

4- I would suggest the authors expand and reconsider the introduction. In addition, I would use the most recent lectures or studies in the related field to add to the references. For example:

https://doi.org/10.1016/j.conbuildmat.2022.129376

https://doi.org/10.1016/j.istruc.2022.07.030

Author Response

First, we would like to thank Reviewer 1 for valuable comments and suggestions. We believe that with his/her help, the overall quality of the manuscript has improved significantly. 

1-The cement mix (in kg/m³ or % of the constituent materials-cement, clay, and ..) was not provided, which implies the impossibility of other authors replicating the study; therefore, I consider it important that these data are reported in the article.

Answer: Thank you for pointing this out. Based on this suggestion, the related section was extended and a detailed sample preparation was added along with the sample composition in wt.%.

2-  Carefully review the language, especially the punctuation, for example, sentence beginnings without capital letters.

Answer: Thank you for the suggestion. The English language was revised thoroughly using a professional proofreading service.

3- Was any specific parameter in cement mortar configuration held constant for comparative purposes?

Answer: Yes, the design of the mixtures followed the same pattern: Mixing of 450 g of the cementious part (OPC; LC3-CL, or LC3-FR), 1350 g of standard sand, and 225 ml of water. This text was added in the section 2.2. with sample preparation.

4- I would suggest the authors expand and reconsider the introduction. In addition, I would use the most recent lectures or studies in the related field to add to the references. For example:

https://doi.org/10.1016/j.conbuildmat.2022.129376

https://doi.org/10.1016/j.istruc.2022.07.030

Answer: Thank you for this suggestion. The firstly mentioned reference was indeed added into Introduction as suggested, nevertheless, the second one deals with single and hybrid-fiber reinforced concrete under drop weight test, which, unfortunately, is not the best fit to our study about effects of alternate wet and dry conditions on the mechanical and physical performance of limestone calcined clay cement mortars immersed in sodium sulfate media.

Reviewer 2 Report

General remarks

 Revise English and technical terms in the following sentences:

·         Line 56: “in wet and dry alternating conditions.” - in alternating wet and dry conditions.

·         Lines 122, 143, 148, 163: “calcined/kept in an oven” - check if “oven” is the correct technical term, I would use furnace instead of oven

·         Table 1. Chemical composition (in mass%) - replace by “in weight%”. Dimension must be indicated for the LOI. Specify the dimension for the residues (I suppose, it is weight%). Replace m2/cc by m2/cm3.

·         Line 126:" The heating was done to remove any moisture, " check if “done” is the correct technical term, and specify, which heating. In the text, only calcination is mentioned as heating, but its aim is not to remove moisture.

·         Lines 144, 164 “until a constant mass was achieved” check if “mass” is the correct technical term (I recommend using weight)

·         Line 168: “the masses of the samples were recorded in gram” - the weight of the samples

·         Lines 171, 173: use weight instead of mass

·         Line 260: “The LC3 samples immersed in a 5% sodium  sulfate  solution  also  recorded  a lower compressive strength” – Revise, a sample could not record any compressive strength

·         Lines 369-371: “The compressive strength test carried out on samples exposed to sulfate attack and the reference samples, we can conclude that LC3 has good resistance to sodium sulfate attack”

·         Line 382: “attained a compressive strength not far off from” – revise not far off, it is not written in the academic style

Remarks to figures

Figs 1-2: Each figure must be clear and understandable on its own without searching for the relevant text. Add the explanation of the labelling system. Furthermore, the standard deviation should be included in the figures.

Figs 3-5: What is the standard deviation of the results? These lines seem to be more likely a stretched square root function then a linear function. Have you tried the fitting of other functions?

Abstract

Lines 18-19: “Each LC3 blend was composed of 50% clinker, 30% calcined clay, 15% lime-stone, and 5% gypsum.”  - The composition of the mixture should be specified (is it in wt%, vol% etc.)

Line 20: Decode OPC as this is the first time of mentioning it.

Line 22: “met the KS-EAS 18-1:2017 requirements” - Specify what this standard applies to.

Introduction

Lines 46-48: “As opposed to clinker, the calcination process does not result in the production of carbon dioxide from the breakdown of the raw materials; hence it helps in the reduction of CO2 emission.” vs Lines 42-43: “Most recently, calcined clay and limestone have been combined as SCMs in cement production” - As I know, the calcination process of limestone can be described as CaCO3 → CaO + CO2. Thus, the statement “calcination process does not result in the production of carbon dioxide from the breakdown of the raw materials” is not valid for the limestone.

 Materials and Methods

Lines 127-128: “50% clinker, 30% of calcined clay, 15% of limestone, and 5% of gypsum.”  - The composition of the mixture should be specified (is it in wt%, vol% etc.)

See notes above for Table 1.

Line 137: “Test samples were prepared in accordance with with KS” - remove a with. Write the basic steps of sample preparation (including the number of the test samples!) as this standard is not available for the readers.

Line 161: “modifications: The cement” - use lower case for the word “the”

Results and Discussion

Lines 180-182: must be deleted, as this is from the template

Lines 195, 199, 202, 228, 273, 282: Portlandite should be written in lower case, as in these cases, it is the name of phase and not a brand or a proper noun

Lines 206-207 “good properties... improved performance” - the wording is general

Line 217: “met the compressive strength requirements of the KS-217EAS 18-1:2017  [17] standard” – specify these requirements

Lines 237-238: “In Na2SO4-H2O; there exist two stable phases; thenardite (Na2SO4),” – remove semicolons

Chapters 3.3.2 and 3.3.3: They have unusual structure, as the figures are right in the beginning of the chapters. Consider re-editing.

 Conclusions

A short summary of the conducted study should be inserted. Add standard deviations to the presented values.

3rd conclusion (lines 380-383): “The designed LC3 cement” – cement is doubled, as LC3 also includes this word.

“still exhibited good stability and attained a compressive strength”- add results for the stability and strength

Line 387: “also met the requirements  of  the aforementioned standard” – specify these requirements

Author Response

First, we would like to thank Reviewer 2 for valuable comments and suggestions. We believe that with his/her help, the overall quality of the manuscript has improved significantly. 

General remarks

Revise English and technical terms in the following sentences:

  • Line 56: “in wet and dry alternating conditions.” - in alternating wet and dry conditions.

Answer: Thank you for pointing this out. The text was modified accordingly.

  • Lines 122, 143, 148, 163: “calcined/kept in an oven” - check if “oven” is the correct technical term, I would use furnace instead of oven

Answer: In our experience, we have used the term an oven in the same context in our previous articles, nevertheless, we have modified the text and replaced “an oven” for the “furnace” as suggested.

  • Table 1. Chemical composition (in mass%) - replace by “in weight%”. Dimension must be indicated for the LOI. Specify the dimension for the residues (I suppose, it is weight%). Replace m2/cc by m2/cm3.

Answer: The text was revised and modified accordingly to this suggestion.

  • Line 126:" The heating was done to remove any moisture, " check if “done” is the correct technical term, and specify, which heating. In the text, only calcination is mentioned as heating, but its aim is not to remove moisture.

Answer: Thank you for pointing this out. The text was rewritten as follows: The drying of the sample was needed to remove any moisture that might have accumulated over time. 

  • Lines 144, 164 “until a constant mass was achieved” check if “mass” is the correct technical term (I recommend using weight)

Answer: Based on this comment, the term “mass” was replaced with “weight” within the manuscript.

  • Line 168: “the masses of the samples were recorded in gram” - the weight of the samples

Answer: The term “mass” was replaced with “weight” within the manuscript.

  • Lines 171, 173: use weight instead of mass

Answer: The term “mass” was replaced with “weight” within the manuscript.

  • Line 260: “The LC3 samples immersed in a 5% sodium sulfate solution also recorded a lower compressive strength” – Revise, a sample could not record any compressive strength

Answer: We agree that this formulation of the sentence is misleading, and thus, the text was rewritten as follows: The LC3 samples immersed in a 5% sodium sulfate solution also had a lower com-pressive strength than the reference LC3 samples that recurred in water.

  • Lines 369-371: “The compressive strength test carried out on samples exposed to sulfate attack and the reference samples, we can conclude that LC3 has good resistance to sodium sulfate attack”

Answer: We agree that something is missing from this sentence, and therefore, the text was rewritten as follows: From the results of compressive strength test carried out on samples exposed to sulfate attack and the reference samples, we can conclude that LC3 has good resistance to sodium sulfate attack.

  • Line 382: “attained a compressive strength not far off from” – revise not far off, it is not written in the academic style

Answer: The related text was modified as follows: “reached similar compressive strength values as the reference LC3 samples”.

Remarks to figures

Figs 1-2: Each figure must be clear and understandable on its own without searching for the relevant text. Add the explanation of the labelling system. Furthermore, the standard deviation should be included in the figures.

Answer: Explanation of the labelling system and Standard deviation bars has been added to Figures 1 and 2.

Figs 3-5: What is the standard deviation of the results? These lines seem to be more likely a stretched square root function then a linear function. Have you tried the fitting of other functions?

Answer: Standard deviation bars were added to the graphs. Unfortunately, we did not try other functions since the ASTM C 1585-13 standard we are using explicitly states that: “The initial rate of water absorption is defined as the slope of the line that is the best fit to, I plotted against the square root of time. Obtain this slope by using least squares, linear regression analysis of the plot of I versus time1/2.”

Abstract

Lines 18-19: “Each LC3 blend was composed of 50% clinker, 30% calcined clay, 15% lime-stone, and 5% gypsum.”  - The composition of the mixture should be specified (is it in wt%, vol% etc.)

Answer: Agreed; the statement has been revised as follows: The composition of each LC3 by weight was 50% clinker, 30% calcined clay, 15% limestone, and 5% gypsum.

Line 20: Decode OPC as this is the first time of mentioning it.

Answer: The definition of OPC was added in the abstract when firstly mentioned.

Line 22: “met the KS-EAS 18-1:2017 requirements” - Specify what this standard applies to.

Answer: The related text was modified as follows: The results showed that LC3 blends met the requirements for KS-EAS 18-1:2017 standard, which specifies the composition and conformity criteria for common cements. The LC3 blends also had a lower rate of initial absorption compared to OPC.

Introduction

Lines 46-48: “As opposed to clinker, the calcination process does not result in the production of carbon dioxide from the breakdown of the raw materials; hence it helps in the reduction of CO2 emission.” vs Lines 42-43: “Most recently, calcined clay and limestone have been combined as SCMs in cement production” - As I know, the calcination process of limestone can be described as CaCO3 → CaO + CO2. Thus, the statement “calcination process does not result in the production of carbon dioxide from the breakdown of the raw materials” is not valid for the limestone.

Answer: We believe that we were misunderstood. Only the raw clays are calcinated, the limestone is used as received, without any heating. The related text was modified as follows for the clarity: “As opposed to clinker, the calcination process does not result in the production of carbon dioxide from the breakdown of the raw clays; hence it helps in the reduction of CO2 emission [5].”

Materials and Methods

Lines 127-128: “50% clinker, 30% of calcined clay, 15% of limestone, and 5% of gypsum.”  - The composition of the mixture should be specified (is it in wt%, vol% etc.)

See notes above for Table 1.

Answer: The text was modified based on this suggestion to read as follows: The LC3 samples were then prepared by mechanical mixing of 50wt.% of clinker, 30wt.% of calcined clay, 15wt.% of limestone, and 5wt.% of gypsum.

Line 137: “Test samples were prepared in accordance with with KS” - remove a with. Write the basic steps of sample preparation (including the number of the test samples!) as this standard is not available for the readers.

Answer: Thanks for taking note, the repeated word “with” deleted. The basic sample preparation steps has been added as advised.

Line 161: “modifications: The cement” - use lower case for the word “the”

Answer: The text was modified based on this suggestion.

Results and Discussion

Lines 180-182: must be deleted, as this is from the template

Answer: Thank you for noticing this mistake. The text was deleted.

Lines 195, 199, 202, 228, 273, 282: Portlandite should be written in lower case, as in these cases, it is the name of phase and not a brand or a proper noun

Answer: the capitalization of portlandite was modified within the manuscript as suggested.

Lines 206-207 “good properties... improved performance” - the wording is general

Answer: The related text was modified as follows: “This makes it possible to maintain good mechanical performance and durability at higher levels of substitution [21, 22].”

Line 217: “met the compressive strength requirements of the KS-217EAS 18-1:2017  [17] standard” – specify these requirements

Answer: The related text was modified and the requirements added as follows:

Both LC3 samples made from clay soil and the ones made from FRCBs met the com-pressive strength requirements of the KS-EAS 18-1:2017 [17] standard. For 2 days, the standard specifies a compressive strength of ≥ 10 MPa for PPC and ≥ 20 MPa for OPC. The LC3-CL and LC3-FR had a 2-day compressive strength of 14.7 MPa and 14.1 MPa respectively. For 7 days, the standard specifies a compressive strength of ≥ 16 MPa for PPC but does not give any value for OPC. The LC3-CL and LC3-FR had a 7-day compressive strength of 32.6 MPa and 31.5 MPa respectively. For 28 days, the standard specifies a compressive strength of ≥ 32.5 MPa for PPC and ≥ 42.5 MPa for OPC. The LC3-CL and LC3-FR had a 28-day compressive strength of 44.0 MPa and 45.4 MPa respectively.

Lines 237-238: “In Na2SO4-H2O; there exist two stable phases; thenardite (Na2SO4),” – remove semicolons

Answer: The semicolons were removed.

Chapters 3.3.2 and 3.3.3: They have unusual structure, as the figures are right in the beginning of the chapters. Consider re-editing.

Answer: Sections 3.3.2 and 3.3.3 re-edited so that figures are not right in the beginning of the chapters

Conclusions

A short summary of the conducted study should be inserted. Add standard deviations to the presented values.

Answer: The short summary of the conducted study was inserted at the beginning of the conclusions.

3rd conclusion (lines 380-383): “The designed LC3 cement” – cement is doubled, as LC3 also includes this word.

Answer: Indeed, the word cement was removed from the text.

“still exhibited good stability and attained a compressive strength”- add results for the stability and strength

Answer: Results added and the statement now reads as below:

The designed LC3 showed good resistance to varying hygrometric conditions. When subjected to alternate wet and dry cycles, the LC3-CL samples attained a compressive strength of 57.7 MPa upon testing. This strength is close to the reference LC3-CL sample which remained fully immersed in water that attained a compressive strength of 63.1 MPa.

Line 387: “also met the requirements of the aforementioned standard” – specify these requirements

Answer: Requirements added and the statement now reads as below:

There is potential for LC3 production in Kenya using clay soil and FRCBs. Chemical analysis showed that the composition of the clay soil and FRCB sampled satisfy requirements for a good pozzolan as prescribed in KS-EAS 18-1:2017 [17] standard. The standard specifies that silicon dioxide (SiO2) content shall not be less than 25% by mass of the material used. From Table 1, it is evident that both clay soil and FRCB samples meet this criterion. The clay soil sample had 39.02 wt.% of SiO2 while FRCB had 43.22 wt.%. The 2, 7, and 28-days compressive strengths also met the standard’s requirements of ≥ 10 MPa for PPC and ≥ 20 MPa for OPC for 2 days, ≥ 16 MPa for PPC for 7 days and ≥ 32.5 MPa for PPC and ≥ 42.5 MPa for OPC for 28 days.

Author Response

First, we would like to thank Reviewer 3 for valuable comments and suggestions. We believe that with his/her help, the overall quality of the manuscript has improved significantly.

COMMENTS AND SUGGESTIONS FOR AUTHORS:

Effects of alternate wet and dry conditions on the mechanical performance of limestone calcined

clay cement mortars immersed in sodium sulfate media.

  1. The absorption (sorptivity test) does not belong to mechanical performance but physical properties, so customize the title.

Answer: Thank you for pointing this out. To satisfy this suggestion, we have updated the title to: Effects of alternate wet and dry conditions on the mechanical and physical performance of limestone calcined clay cement mortars immersed in sodium sulfate media.

  1. Compressive strength tests were carried out for the age 2, 7, 28 days. Why is there statement “LC3 blends exhibited higher compressive strength than the OPC samples at 84 days of curing” in abstract. The methodology used must be clearly explained in the paper.

Answer: Thank you for this question. Indeed, this statement can be confusing. We have revised the text portion to read as follows for more clarity:

OPC showed higher compressive strength than LC3 blends for testing ages 2,7, and 28 days. However, the LC3 samples utilized in the sodium sulfate attack experiment, which were later tested after 84 days, exhibited higher compressive strengths than OPC tested after the same period.

  1. Some references are outdated. It is recommended to use recent references in the last 5 – 10 years.

Answer: All used references are for our study essential, and in our opinion, the oldest one from 1998, referring to the Lea’s chemistry, is still very valuable for the current research.

  1. This paper is not pertaining to MICP as concrete self-healing, so ref. no. 1 should be removed.

Answer: We have removed ref no: 1. and used this one instead:

Habert, G.; Miller, S.A.; John, V.M.; Provis J.L.; Favier A.; Horvath A.; Scrivener, K.L. “Environmental impacts and decar-bonization strategies in the cement and concrete industries,” Nat. Rev. Earth Environ., vol. 1, no. 11, pp. 559–573, Sep. 2020, doi: 10.1038/s43017-020-0093-3.

  1. “Commercially obtained PPC conforming to 32.5 MPa and OPC conforming to 42.5 MPa were used as reference cements” but the compressive strength obtained at the age of 28 days is far above it, namely 48.40 MPa for OPC and 41.35 MPa for PPC. Explain how much strength target is desired. Mix designs for all variations of cement mortars must be absolutely presented.

Answer: Thank you for pointing this out. We updated the related text, as suggested. More specifically, from section 2.3.1, it is now evident that the w/c used is 0.5 in the mix design. The higher strengths attained at the age of 28 days for PPC, and OPC has to do with the quality of cement being manufactured. The KS EAS 148-1:2017 standard used for cement in Kenya specifies the following strength limits at 28 days; PPC ≥ 32.5, ≤ 52.5; OPC ≥ 42.5, ≤ 62.5. The 32.5 and 42.5 are just the minimum values allowed but not the upper ceiling. The 42.5 for PPC and 48.40 for OPC are well within the standard’s range.

  1. Physical properties of SCMs used in addition to chemical composition, as fineness, bulk density, specific density must be presented to thoroughly analyze its effect on the compressive strength.

Answer: The amount of particles retained in a standard 45-micron sieve is presented in Table 1. We believe this is a good indication of the fineness of the SCM and cement particles. The values obtained met the  KS EAS 148-1:2017 standard specification for pozzolan as well as ASTM C618 - Standard Specification for Coal Fly Ash and Raw or Calcined Natural Pozzolan for use in Concrete.  At the moment, we do not have data on specific or bulk density. We believe that the provided physical and chemical properties are sufficient to analyze the effects of sulfate attack on cement.

  1. The main hydration process involves all clinkers in cement, namely C2S, C3S, C3A and C4AF as well as gypsum CaSO4.H2O which reacts with water (H2O) as a reactant to form C-S-H and CH (CaOH2) phases, in which C-S-H is responsible for the establishment of concrete strength. Equations 2 to 5 must be formulated in full, for example, the water as reactant must be written H2O and so on. Chemical reactions must meet the principle of balance.

Answer: Thank you for pointing this out. The related text (equations) was modified as suggested.

  1. The lost of compressive strength after subjecting to wet – dry cycles must be displayed in table or diagram to be compared each other to find out how significant the reduction is.

Answer: Thank you for this interesting suggestion. A table showing the loss in strength after wet-dry cycle test has been added; please see Table 3.

  1. Originality and relevancy of paper are average.
  2. Consistency of title with content is average.
  3. Consistency of conclusion with content is good.

Round 2

Reviewer 3 Report

There is typo in Fig. 1 (made instead of "dade").